# Impact of clinical pathways on enhancing compliance with evidence-based therapies for Heart failure with reduced ejection fraction– A retrospective cohort study

Naga Sasidhar Kanaparthy[1,2,3]*, Stephen Possick[4,5‡], Wei Teng[2], Christopher Maulion[4,5‡], Nancy Kim[5,6]

1 Department of Emergency Medicine, Yale University School of Medicine, New Haven, Connecticut, United States of America, 2 Biomedical Informatics and Data Science, Yale University School of Medicine, New Haven, Connecticut, United States of America, 3 Clinical Informatics, Veteran Health Administration, West Haven, Connecticut, United States of America, 4 Department of Cardiology, Yale University School of Medicine, New Haven, Connecticut, United States of America, 5 Department of Cardiology, Yale New Haven Health System, Yale New Haven Hospital, New Haven, Connecticut, United States of America, 6 Section of General Internal Medicine, Yale University School of Medicine, New Haven, Connecticut, United States of America

☺ These authors contributed equally to this work.
‡ These authors also contributed equally to this work.
* Naga.kanaparthy@yale.edu

## Abstract

### Background

Heart failure (HF) mortality is rising despite robust evidence-based guidelines. Hospitalization presents an opportune time to optimize care. Inpatient care pathways (CP) embedded in the electronic health record (EHR) can enhance adherence to guidelines by providing real-time decision support.

### Objective

To measure the impact of the heart failure reduced ejection fraction (HFrEF) CP on early diuretic use, goal-directed medical therapy (GDMT), and referral to comprehensive HF management (CHFM) on discharge.

### Methods

A HFrEF CP embedded into the EHR was built by multidisciplinary providers across a large academic medical center. Providers could place orders and document notes directly from the pathway. We conducted a retrospective cohort study of all the HF hospitalizations two years after pathway implementation and further limited the cohort to those with ejection fraction (EF) ≤ 40%. We compared rates of early diuretic use, GDMT, and referral to CHFM between cohorts defined by pathway use.

**Data availability statement:** The data has been uploaded to a public data repository (DOI): https://doi.org/10.5061/dryad.h18931zz9.

**Funding:** The author(s) received no specific funding for this work.

**Competing interests:** No.

## Results

1,639 patients contributed 2,062 hospitalizations. The pathway was opened in 362 (17.6%) hospitalizations. There were no significant clinical or sociodemographic differences between groups. Bivariate analyses revealed more GDMT ($\chi^2$ (1) = 3.651, p = 0.056); while multivariable analysis showed more early diuretics (OR = 1.58, 95% CI 1.19, 2.09, p = 0.002) and referrals to CHFM (OR = 3.24, 95% CI 2.54, 4.12, p < 0.001) in the pathway group.

## Conclusion

Despite low utilization, pathway use was associated with more early diuretics and referrals to CHFM with a trend toward more GDMT on discharge. CPs are a feasible strategy to optimize care for hospitalized HF patients.

## Introduction

Congestive heart failure (HF) accounts for approximately 1.2 million US hospitalizations annually with direct cost of treating patients over $30 billon [1]. Although the American Heart association (AHA) has published best practice guidelines with detailed instructions about medications, weight management, smoking cessation, follow up appointments and other critical post-discharge follow-up; many patients never benefit from evidence-based practice due to non-compliance with these validated recommendations [2–4]. This may be a key reason that HF admissions and mortality are rising [5].

One important guideline that enhances survival and improves quality of life for HF with reduced EF (HFrEF) patients is adherence to goal-directed medical therapy (GDMT) as evidenced by the STRONG-HF trial [6,7]. Bassi et al. demonstrated that despite a 73% reduction in mortality risk over two years with GDMT, its use is not widespread [8]. GDMT remains challenging in clinical practice due to polypharmacy, patient comorbidities, and care fragmentation. Hospitalization presents a crucial window to initiate comprehensive goal-directed medical therapy (GDMT) with all four agents in heart failure patients, ensuring optimal treatment while under close clinical supervision and addressing the real-world gap of suboptimal GDMT prescription in the ambulatory setting.

Similarly, in the inpatient setting, symptom control with loop diuretics is the mainstay for patients with HF, however early and aggressive diuresis is underachieved [9]. Rubio-Garica et al. measured residual congestion after discharge and noted a third of patients still had congestion after seven days [10]. One barrier contributing to this care gap includes clinician knowledge and behavior [11–13]. With seven million articles published annually, clinicians struggle to keep pace with new evidence [14].

Care pathways (CP) represent one strategy to enhance consistency of care and foster adherence to best practice for patients with multiple complex medical issues [12,15]. CPs at their core represent multi-disciplinary plans that provide real-time

decision support by integrating guidelines and current evidence at the point of care when the clinician needs it. Additionally, local workflow can be expedited when CPs are seamlessly integrated with the electronic health record (EHR), which allows physicians to perform supported tasks, place orders, and document notes in one sitting. CPs differ from other efforts like interruptive and non-interruptive alerts, discharge checklists, patient reminders, and physician education [15], which like isolated EHR alerts, have not proved effective [12]. CPs can be updated regularly to offer the most recent guidelines around the clock for providers, thereby expediting the time for evidence-based discovery to reach the bedside.

The aim of this study was to evaluate the impact of a comprehensive CP for HFrEF embedded in the EHR across a large healthcare system. We hypothesized that the use of the HFrEF CP pathway would increase early diuretic therapy, GDMT and ambulatory referrals to comprehensive HF management (CHFM) on discharge, an important intervention associated with reduced hospital readmission [16,17].

## Methods

### Ethical approval

This study was approved by the Institutional Review Board of Yale University who granted a waiver of informed consent (Protocol #2000032391). For this study, the data was retrieved on December 10th, 2023. The authors did not have access to information that could identify individual participants during or after data collection, except for Dr. Wei Teng, who had access to the information at the time of data retrieval from database, but did not have access to it thereafter.

### Design, setting, patients

The Yale-New Haven Health System (YNHHS) is a large healthcare system with five delivery networks comprised of seven hospitals in Connecticut and Rhode Island, ranging from small community hospitals to large tertiary care academic centers. We conducted a retrospective cohort study of all inpatient adults aged 18 or older who were hospitalized at YNHHS for a coded primary discharge diagnosis of HF from October 1, 2021, to September 30, 2023. From there we limited our sample to those patients who had echocardiographic evidence of an ejection fraction ≤ 40% one year prior to or during their index admission for HF. We defined the cohort using International Classification of Disease, Tenth Revision, Clinical Modification (ICD-10-CM) codes in accordance with methods used by the Centers for Medicare & Medicaid Services (List of HF ICD-10 diagnosis codes are shown in S2 File). Unique individuals could have contributed to more than one HF hospitalization.

### Intervention–The care signature pathway for HFrEF

A HFrEF clinical care pathway for inpatient care was developed by local consensus from multi-disciplinary stakeholders across YNHHS including Hospital Medicine, Cardiology, Nursing, Pharmacy, Physical Therapy, Care Management, and Social Work (**Figures:** Screenshots of admission (Fig 1) and discharge (Fig 2) pathways). The full pathway can be found in the S1 File. Its components were derived from existing national guidelines as well as input from front-line clinicians regarding their work-flow and point-of care decision-making. This pathway was considered our "Care Signature" and was embedded into the EHR on September 14, 2021, at all seven hospitals across YNHHS. The pathway was developed using proprietary software from AgileMD, Inc. (San Francisco, CA) and socialized with YNHHS providers through various mechanisms including e-mail communication, newsletters, dedicated educational conferences, ad-hoc meetings, and word-of-mouth. We collected data for the 24 months after the pathway was published.

The pathway provided clinical decision-support at the point of care and allowed providers to document and place orders directly from the pathway. Because of the complex nature of HF diagnosis, management, and follow-up care, the HFrEF pathway was organized into "tabs" that corresponded to actions necessary on admission, in subsequent hospital days, and on discharge. Guidance on admission included confirming a HF diagnosis with particular attention to early

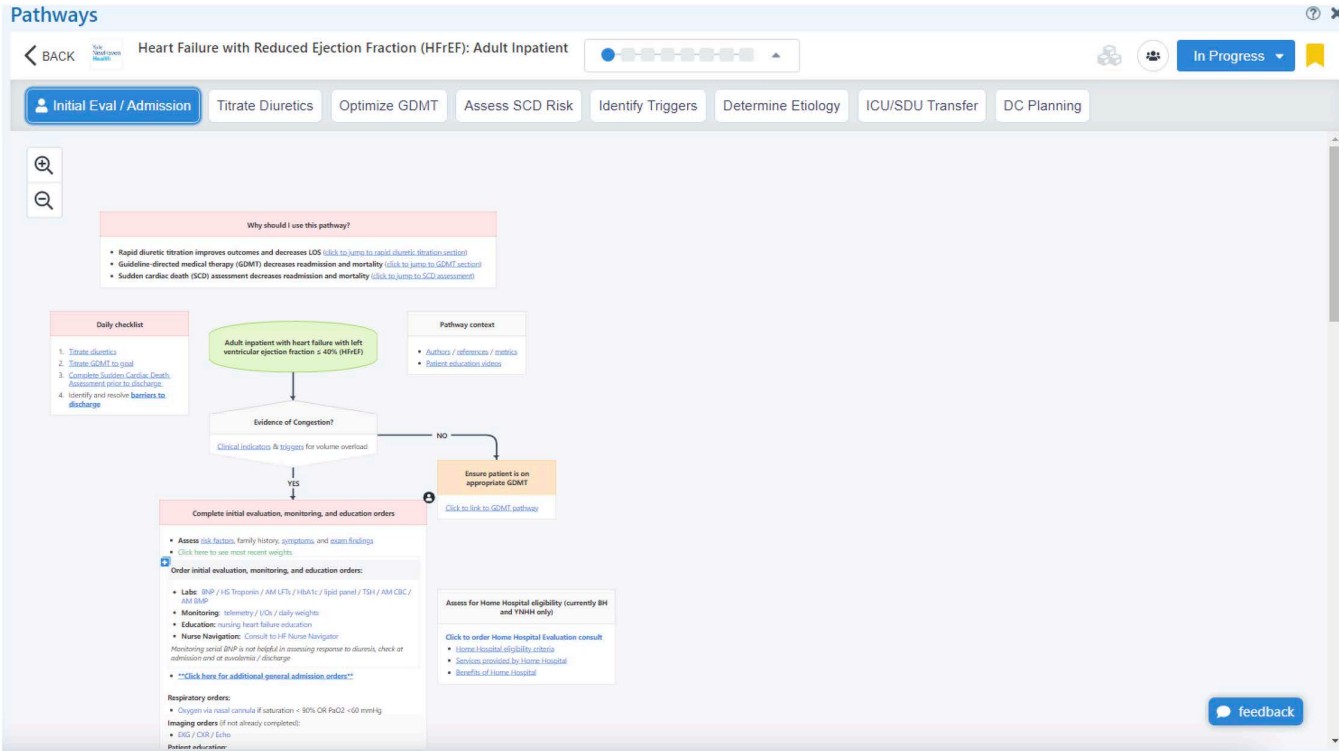

**Fig 1. HFrEF care signature pathway: Admission tab.**

aggressive diuretic therapy. In subsequent days, there are timely reminders to initiate and titrate GDMT and reminders to clinicians to place ambulatory referrals to CHFM on discharge. To be consistent with current guidelines, we defined GDMT as the prescription of all four classes of drugs on discharge: angiotensin receptor-neprilysin inhibitors or angiotensin-converting enzyme inhibitors or angiotensin (II) receptor blockers (ACEI/ARB/ARNI); beta blocker (BB); sodium-glucose cotransporter-2 inhibitors (SGLT2i); and mineralocorticoid receptor antagonist (MRA) [4].

Pathways had to be activated voluntarily by providers, and were available to all providers. There was no forced work-flow to mandate clinicians use the pathway. All actions available on the pathway were also available through standard order entry off the pathway. While the pathway was equally accessible, its voluntary nature introduces the possibility of self-selection bias, as providers who chose to use the pathway may differ in their clinical practices or engagement with guideline-based care. However, we anticipate minimal bias in outcome measurement, as all relevant data were extracted directly from the EHR using standardized criteria.

## Outcomes

### Utilization and outcomes

Because providers had to enter the pathway intentionally, pathway utilization was measured. Utilization was defined as any opening of any component of the HFrEF pathway during hospitalization. Pathways were designed to improve the process measures felt to be key drivers of clinical outcomes. Therefore, the primary outcomes of the study were early diuretic therapy defined as any diuretic given within eight hours of admission, GDMT on discharge, and ambulatory referral to CHFM on discharge.

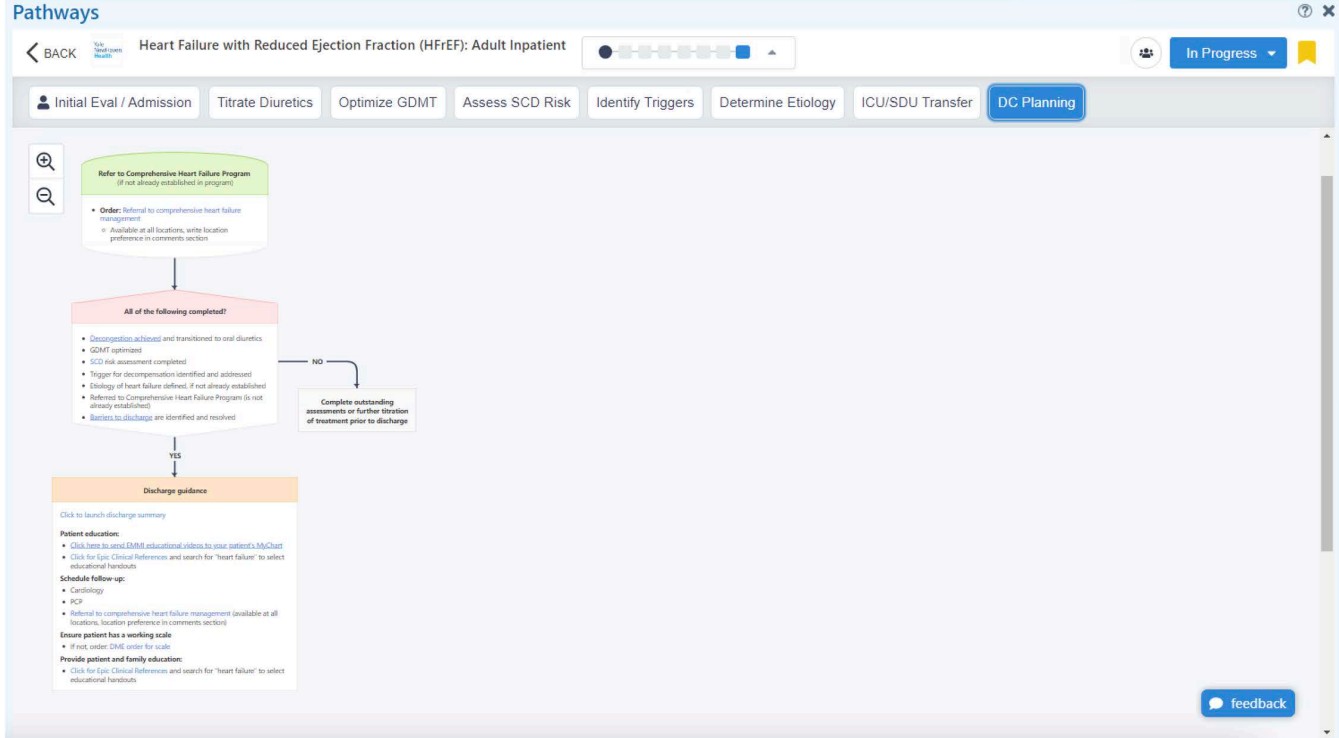

**Fig 2. HFrEF care signature pathway: Discharge tab.**

## Statistical analysis

Sociodemographic characteristics included age, sex, race/ethnicity, insurance status, and current tobacco use. Asian/ Pacific Islander and American Indian/Alaska Native were grouped in the "Other" category because of small numbers. We analyzed race/ethnicity, and insurance status for any disparities in pathway use related to these factors. The Charlson co-morbidity index was assigned based on ICD-10 secondary diagnosis codes. Descriptive statistics are presented as median (interquartile range) for continuous variables and proportions for categorical variables.

Multivariable logistic regression models were constructed to determine the contribution of HFrEF pathway use on the primary outcomes that reaches statistical significance in bivariate analyses. Logistic regression models estimate the odds of an outcome occurring based on one or more predictor variables. Statistical analyses were conducted in IBM SPSS statistics, Version 26.0 (IBM Corp., Armonk, NY) and p ≤ 0.05 was considered statistically significant for all outcomes.

## Results

### Pathway use and patient characteristics

1,639 unique patients contributed a total of 2,062 hospitalizations, following the exclusion of 92 hospitalizations due to inpatient death. The HFrEF pathway was opened in 362 (17.6%) of hospitalizations. For all HF hospitalizations, the median age was 71 years; 1,318 (63.9%) were male; 1,154 (56%) were Non-Hispanic White; 626 (30.4%) were non-Hispanic Black; 220 (10.7%) were Hispanic; and 62 (3%) were classified as Other. Primary insurance types in the full cohort were private (49%), Medicare (33.7%), and Medicaid (14.9%) There were no statistically significant differences regarding demographics or primary insurance between those hospitalizations with versus without pathway use (**Table**

**Table 1. Patient characteristics.**

| | Full Cohort | Pathway Used | Pathway Not Used | P value |
|---|---|---|---|---|
| **Hospitalizations,** n (%) | 2,062 | 362 (17.6) | 1,700 (82.4) | N/A |
| **Demographics** | | | | |
| Age in years, median (IQR) | 71 (61,82) | 71 (60,83) | 71 (62,82) | N/A |
| Male, n (%) | 1,318 (63.9) | 239 (66.0) | 1,079 (63.5) | 0.359 |
| **Race/ethnicity,** n (%) | | | | 0.728 |
| Non-Hispanic white | 1,154 (56.0) | 211 (58.3) | 943 (55.5) | |
| Non-Hispanic Black | 626 (30.4) | 101 (27.9) | 525 (30.9) | |
| Hispanic | 220 (10.7) | 39 (10.8) | 181 (10.6) | |
| Other | 62 (3.0) | 11 (3.0) | 51 (3.0) | |
| **Primary insurance,** n (%) | | | | 0.185 |
| Private | 1,010 (49.0) | 181 (50.0) | 829 (48.8) | |
| Medicare | 695 (33.7) | 107 (29.6) | 588 (34.6) | |
| Medicaid | 308 (14.9) | 64 (17.7) | 244 (14.4) | |
| Other | 49 (2.4) | 10 (2.8) | 39 (2.3) | |
| **Tobacco use,** n (%) | | | | 0.800 |
| Current smoker | 270 (13.1) | 50 (13.8) | 220 (12.9) | |
| Past smoker | 66 (3.2) | 13 (3.6) | 53 (3.1) | |
| Never smoker | 1726 (83.7) | 299 (82.6) | 1,427 (83.9) | |
| **Charlson Comorbidity Index,** median (IQR) | 4 (3,6) | 4 (3,6) | 3 (2,5) | |
| **Ejection fraction,** median (IQR) | 26 (17,32) | 26 (17,33) | 27(18,32) | N/A |
| **Comorbidities,** n (%) | | | | |
| COPD | 349 (16.9) | 58 (16) | 291 (17.1) | 0.614 |
| Chronic kidney disease | 729 (35.4) | 119 (32.9) | 610 (35.9) | 0.277 |
| Hypertension | 1173 (56.9) | 205 (56.6) | 968 (56.9) | 0.914 |
| Diabetes | 652 (31.6) | 117 (32.3) | 535 (31.5) | 0.752 |
| Obesity | 648 (31.4) | 125 (34.5) | 523 (30.8) | 0.161 |
| Peripheral vascular disease | 343 (16.6) | 66 (7.2) | 277 (16.3) | 0.369 |
| Malnutrition or BMI<20 | 149 (7.2) | 26(7.2) | 123(7.2) | 0.972 |
| **Hospital admissions in prior 12 months,** n (%) | | | | <0.001 |
| Zero or 1 | 959 (46.5) | 200 (55.3) | 759 (44.7) | |
| 2 | 464 (22.5) | 77 (21.3) | 387 (22.8) | |
| >/=3 | 639 (31.0) | 85 (23.5) | 554 (32.6) | |

1). Patients for whom the HF pathway was accessed did not differ from non-pathway patients in individual comorbidities, Charlson Comorbidity Index, or Ejection Fraction. Significant differences were observed, however, in prior hospital admissions among patients on and off the pathway.

## Association between pathway use and outcomes

In bivariate analysis, the use of diuretics within the first 8 hours of hospitalization was higher among hospitalizations with pathway use (80.1% vs 71.2%, p<0.001). Intensive care unit use was lower in the pathway group (14.6% vs 20.6%, p=0.010). (Table 2). Referrals to CHFM were more frequent among pathway users (51.4% vs 24.2%, p<0.001). The use of ACE/ARB/ARNI at discharge was higher (75.1% vs 65.6%, p<0.001) for patients on pathway and there was a higher

**Table 2. Characteristics during hospitalization.**

| | Full Cohort | Pathway Used | Pathway Not Used | P value |
|---|---|---|---|---|
| Diuretics first 8 hours of admission, n (%) | 1,500 (72.7) | 290 (80.1) | 1,210 (71.2) | <0.001 |
| Non-invasive ventilation, n (%) | 404 (19.6) | 61 (16.9) | 343 (20.2) | 0.142 |
| ICU, n (%) | 403 (19.5) | 53 (14.6) | 350 (20.6) | 0.010 |

proportion of GDMT on discharge in the pathway group (27.9% vs 23.2%, p=0.056), although this did not meet statistical significance (**Table 3**).

In multivariable analysis, admissions where the pathway was used had greater odds of early diuretic use in the first eight hours of admission when compared to no pathway use (OR = 1.58, 95% CI 1.19–2.09, p=0.002). Other factors associated with increased odds of receiving early diuretics included: age, prior admissions in the last 12 months, past smoking status, Charlson Comorbidity Index, and hypertension (**Table 4**). Pathway use was also associated with greater odds of ambulatory referral to CHFM (OR = 3.24, 95% CI 2.54–4.12, p<0.001). Other factors associated with greater odds of ambulatory referral to CHFM included: age, Black and Hispanic race/ethnicity, Charlson Comorbidity Index, and use of non-invasive ventilation during hospitalization (**Table 5**).

## Discussion

This study highlights the feasibility of implementing a HFrEF inpatient care pathway in the EHR across a large healthcare system. It demonstrates that clinicians' pathway utilization during hospitalizations is associated with more patients receiving early diuretics, ambulatory referrals to CHFM, and a non-statistical trend toward GDMT at discharge.

Diuretics remain the mainstay in inpatient management of HF, yet in most cases, patients do not receive early or aggressive diuretics which can lead to longer length of stay [18]. Despite the low utilization of the pathway, we noted that the pathway's association with early diuretic therapy remained statistically significant in multivariable analysis. Because measuring the amount of diuretic therapy accurately was beyond the scope of the paper and the pathway directed toward time and dose in one order, we chose the eight-hour diuretic as a marker for early aggressive therapy. The HFrEF pathway not only specified how to titrate initial dose of diuretics on admission relative to the patient's stable home dose; it

**Table 3. Discharge outcomes.**

| | Full Cohort | Pathway Used | Pathway Not Used | P value |
|---|---|---|---|---|
| **GDMT at discharge,** n (%) | 495 (24.0) | 101 (27.9) | 394 (23.2) | 0.056 |
| ACE or ARB or ARNI | 1,388 (67.3) | 272 (75.1) | 1,116 (65.6) | <0.001 |
| Beta Blocker | 1,660 (80.5) | 303 (83.7) | 1,357 (79.8) | 0.091 |
| SGLT-2 inhibitor | 1,054 (51.1) | 198 (54.7) | 856 (50.4) | 0.133 |
| Mineralocorticoid receptor antagonist | 890 (43.2) | 170 (47.0) | 720 (42.4) | 0.108 |
| **Referral to comprehensive HF management,** n (%) | 598 (29.0) | 186 (51.4) | 412 (24.2) | <0.001 |
| **Discharge disposition,** n (%) | | | | 0.022 |
| Home without services | 883 (42.8) | 173 (47.8) | 710 (41.3) | |
| Home with services | 661 (32.1) | 94 (26.0) | 567 (33.4) | |
| SNF | 331 (16.1) | 63 (17.4) | 268 (15.8) | |
| Hospice | 74 (3.6) | 8 (2.2) | 66 (3.9) | |
| Other | 113 (5.5) | 24 (6.6) | 89 (5.2) | |
| **30-Day Readmission Rate,** n (%) | 452 (21.9) | 74 (20.4) | 378 (22.2) | 0.045 |

**Table 4. Early diuretic administration multivariable analysis.**

| | Odds Ratio | 95% Confidence Interval | | P value |
|---|---|---|---|---|
| **Pathway Used** | 1.58 | 1.19 | 2.09 | 0.002 |
| **Demographics** | | | | |
| Age | 1.02 | 1.01 | 1.02 | <.001 |
| Gender | 1.07 | 0.86 | 1.32 | 0.552 |
| **Race** | | | | |
| Non-Hispanic White | Reference | | | |
| Non-Hispanic Black | 1.14 | 0.89 | 1.46 | 0.291 |
| Hispanic | 1.13 | 0.81 | 1.58 | 0.480 |
| Other | 0.99 | 0.56 | 1.76 | 0.972 |
| **Admission in the last 12 months** | 1.20 | 1.06 | 1.36 | 0.004 |
| **Smoking Status** | | | | |
| Never smoker | Reference | | | |
| Current smoker | 1.28 | 0.72 | 2.28 | 0.400 |
| Past smoker | 1.35 | 0.98 | 1.86 | 0.065 |
| **Clinical Characteristics** | | | | |
| Hypertension | 1.27 | 1.03 | 1.55 | 0.023 |
| Charlson Comorbidity Index | 0.95 | 0.91 | 0.99 | 0.016 |

**Table 5. Referral to comprehensive heart failure management multivariable analysis.**

| | Odds Ratio | 95% Confidence Interval | | P value |
|---|---|---|---|---|
| **Pathway Used** | 3.24 | 2.54 | 4.12 | <.001 |
| **Demographics** | | | | |
| Age | 0.99 | 0.98 | 1.00 | 0.002 |
| Gender | 0.99 | 0.80 | 1.22 | 0.902 |
| **Race** | | | | |
| Non-Hispanic White | Reference | | | |
| Non-Hispanic Black | 1.30 | 1.01 | 1.67 | 0.038 |
| Hispanic | 1.43 | 1.02 | 1.99 | 0.035 |
| Other | 1.22 | 0.68 | 2.16 | 0.508 |
| **Primary Insurance** | | | | |
| Medicare | Reference | | | |
| Medicaid | 1.12 | 0.80 | 1.58 | 0.498 |
| Private | 0.97 | 0.77 | 1.22 | 0.777 |
| Other | 0.77 | 0.38 | 1.58 | 0.481 |
| **Admission in the last 12 Months** | 1.13 | 1.00 | 1.29 | 0.056 |
| **Clinical Characteristics** | | | | |
| Hypertension | 1.08 | 0.88 | 1.33 | 0.462 |
| Charlson Comorbidity Index | 0.95 | 0.90 | 0.99 | 0.021 |
| Non-Invasive Ventilation | 0.55 | 0.42 | 0.73 | <.001 |

allowed clinicians to place the order directly from the pathway without interruption to their workflow. This may be the mechanism by which the pathway exerted its association with early diuretics.

GDMT is defined as the prescription of all four drug classes: ACEI/ARB/ARNI, BB, SGLT2i, and MRA [4]. The pathway is explicit about each of these medications as well as the guidance that a small dose of all four medications is preferred to larger doses of one to three of these medications. In this way, the pathway redresses any knowledge gaps that may lead to non-achievement of GDMT on discharge. GDMT is one of the most important factors in decreasing HF mortality, yet national rates of four-drug GDMT remain unknown. The rates at our institution of 23–28% are consistent with the general trends reported in literature for three-drug GDMT such as the EVOLUTION HF study [19]. The pathway is a promising intervention to remind clinicians to prescribe all four classes of drugs upon hospital discharge which experts cite as a prime opportunity to begin these medications safely [7].

Although complete GDMT at discharge did not reach statistical significance—likely reflecting existing prescribing habits and limited integration time—the observed positive trend remains clinically important given GDMT's known survival benefits. Notably, the significant improvement in ACEI/ARB/ARNI use is particularly impactful due to their well-established benefits on survival and hospitalization rates.

The statistically significant increase in early diuretic use is clinically valuable, as timely diuretics improve symptoms and shorten hospital stays. Similarly, increased referrals to comprehensive HF management substantially enhance care continuity, reduce readmissions, and improve patient outcomes.

Prompt ambulatory follow-up within 14 days of discharge is associated with reduced readmissions and mortality [16,17,20]. Moreover, a multidisciplinary strategy including utilization of dedicated CHFM clinics has been shown in multiple studies to decrease rehospitalizations and mortality [21]. Our dedicated HF clinics provide a setting for extensive patient and caregiver education regarding HF and its treatment. These HF clinics also serve as a hub for expedited referral to other critical cardiovascular services including electrophysiology, valve-specialty clinics, and advanced therapies when appropriate for patients with progressive disease. Patients for whom the pathway was used, were three times more likely to receive a referral. To our knowledge this is the only intervention to yield such results. Given that the pathway was embedded in the EHR and did not require extra resources such as a dedicated navigator or coordinator to refer patients, the pathway is a feasible intervention to optimize connections to such programs. As such, the pathway likely fills a knowledge and operational gap among clinicians who may not know that such programs exist or how to access this resource in our large healthcare system.

Availability at any time is a hallmark of our Care Signature pathways which are always accessible in the EHR. They are self-contained in that they do not require a nurse navigator or consultant's participation in their deployment. They are seamlessly integrated in the patient's chart and are visible regardless of care setting such as the Emergency Department, general floors, or ICU. They differ from individual interruptive alerts (e.g.,: "Best Practice Alerts" in Epic) which are frequently disruptive, dismissed, and complicated by alert fatigue [12]. Furthermore, such alerts "pop up" at algorithmically prescribed moments rather than provide fluid clinical decision support (CDS) when needed by providers. A more comprehensive approach such as the CP better aligns with the five principles of effective CDS: delivering the right information, to the right person, in the right format, through the right channel at the right time within the workflow [22]. CPs are meticulously designed to encompass all facets of care, providing a structured template for healthcare providers to ensure optimal care delivery across all phases (refer to: Fig 1) and are immune to changes in doctors, nurses, or case managers during a hospitalization. The existing literature around CPs and HF is limited in both volume and age. We found a small number of papers, most of which were over 12 years old and did not address EHR integration [13]. The most recent evidence comes from European hospitals, where the introduction of care pathways during the pandemic showed improvements in outcomes such as prescription adherence and post-discharge follow-up [23,24]. That said, our findings are generally consistent with prior literature, suggesting that CPs are effective in increasing prescriptions of individual medications in HF. Our study adds to this body of knowledge and demonstrate that EHR embedded CPs may unlock the potential for

multiple improved process and clinical outcomes in HF. Building on prior work such as the Program to Optimize Heart Failure Patient Pathways (PRO-HF), which focused on outpatient care, our study expands the evidence base by evaluating an inpatient, EHR-integrated pathway across a large, diverse health system. This distinction underscores the potential for such pathways to standardize care and improve outcomes in the inpatient setting [25].

If adapted and individualized to local clinical workflows and institutional needs, similar care pathways could yield meaningful improvements in guideline adherence and patient outcomes across diverse healthcare settings.

## Limitations

Our study has several limitations. First, the pathway's low utilization may reflect a common challenge in implementing clinical decision support tools [26]. In our healthcare system, most HF patients are managed by general hospitalists who, due to the routine nature of HF admissions, may not perceive the need to consult a pathway. This perception can lead to underutilization, despite the complexity and evolving nature of HF guidelines, which span over 40 pages [4].

Second, there is potential for selection bias, as physicians who chose to use the pathway may differ systematically from those who did not, possibly due to greater engagement with evidence-based care or differing clinical practices. This self-selection could influence the observed associations between pathway use and improved outcomes.

Third, while our healthcare system serves a diverse patient population, the generalizability of our findings to other settings is uncertain. Differences in healthcare infrastructure, provider roles, and patient demographics may affect the applicability of our results elsewhere.

## Conclusion

Implementation of an inpatient HF care pathway is feasible, and despite low utilization leads to improvements in early diuretics and referral to CHFM, which has been shown to improve outcomes in patients with HF. CPs also have the potential to increase the number of patients receiving both individual components as well as complete GDMT, a critical mediator of reduced HF mortality. This study contributes to the field by demonstrating that EHR-embedded CPs can effectively standardize evidence-based practices within complex healthcare systems. Future research should focus on identifying barriers to broader pathway adoption, evaluating clinical outcomes in diverse settings, and developing tailored implementation strategies to enhance clinician engagement and utilization.

## Supporting information

**S1 File. Supplement 1 provides the screen shots of the entire existing care signature pathway at Yale New Haven Hospital.**
(PDF)

**S2 File. Supplement 2: List of Heart failure diagnosis codes from ICD-10.**
(DOCX)

**S3 File. Description** *of dataset structure, variable definitions, and data sources used in the analysis of HFrEF clinical pathway compliance.*
(TXT)

**S4 File. De-***identified raw patient-level data extracted from EHR, used to evaluate adherence to guideline-directed therapies for HFrEF.*
(CSV)

**S5 File. *STROBE checklist.***
(DOC)

## Author contributions

**Conceptualization:** Naga Sasidhar Kanaparthy, Stephen Possick, Nancy Kim.

**Data curation:** Wei Teng, Nancy Kim.

**Formal analysis:** Naga Sasidhar Kanaparthy, Wei Teng.

**Investigation:** Naga Sasidhar Kanaparthy, Nancy Kim.

**Methodology:** Naga Sasidhar Kanaparthy, Stephen Possick, Christopher Maulion, Nancy Kim.

**Supervision:** Nancy Kim.

**Visualization:** Christopher Maulion.

**Writing – original draft:** Naga Sasidhar Kanaparthy, Nancy Kim.

**Writing – review & editing:** Naga Sasidhar Kanaparthy, Stephen Possick, Wei Teng, Christopher Maulion, Nancy Kim.

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
