## [Decision Letter · Decision Letter 0]

16 Apr 2025

Dear Dr. Kanaparthy,

**ACADEMIC EDITOR**

The reviewers in general have had trouble accessing your tables and figures. Please address this together with all pending comments.

We look forward to receiving your revised manuscript.

Kind regards,

Gbolahan Olatunji, M.D.

Academic Editor

PLOS ONE

Journal Requirements:

1. Please ensure that your manuscript meets PLOS ONE's style requirements, including those for file naming. The PLOS ONE style templates can be found at https://journals.plos.org/plosone/s/file id=wjVg/PLOSOne_formatting_sample_main_body.pdf and https://journals.plos.org/plosone/s/file?id=ba62/PLOSOne_formatting_sample_title_authors_affiliations.pdf

no

5. In the online submission form, you indicated that data are available from the Yale Univeristy Institutional Data Access / Ethics Committee for researchers who meet the criteria for access to confidential data.

6. Please remove all personal information, ensure that the data shared are in accordance with participant consent, and re-upload a fully anonymized data set.

Reviewers' comments:

Reviewer's Responses to Questions

**Comments to the Author**

1. Is the manuscript technically sound, and do the data support the conclusions?

Reviewer #1: Partly

Reviewer #2: Yes

Reviewer #3: Yes

Reviewer #4: Partly

Reviewer #5: Yes

Reviewer #6: Yes

2. Has the statistical analysis been performed appropriately and rigorously?

Reviewer #1: I Don't Know

Reviewer #2: Yes

Reviewer #3: Yes

Reviewer #4: No

Reviewer #5: Yes

Reviewer #6: Yes

3. Have the authors made all data underlying the findings in their manuscript fully available?

Reviewer #1: No

Reviewer #2: Yes

Reviewer #3: No

Reviewer #4: No

Reviewer #5: Yes

Reviewer #6: Yes

4. Is the manuscript presented in an intelligible fashion and written in standard English?

Reviewer #1: Yes

Reviewer #2: Yes

Reviewer #3: Yes

Reviewer #4: No

Reviewer #5: Yes

Reviewer #6: Yes

Reviewer #1: I cannot provide details for the present manuscript if I don't have the tables. I apologize for the incovenience.

o be accepted for publication in PLOS ONE, research articles must satisfy the following criteria:

1. The study presents the results of original research.

2. Results reported have not been published elsewhere.

3. Experiments, statistics, and other analyses are performed to a high technical standard and are described in sufficient detail.

4. Conclusions are presented in an appropriate fashion and are supported by the data.

5. The article is presented in an intelligible fashion and is written in standard English.

6. The research meets all applicable standards for the ethics of experimentation and research integrity.

7. The article adheres to appropriate reporting guidelines and community standards for data availability.

Therefore, your evaluation of this submission and your recommendation to the Academic Editor should focus on the scientific soundness of the work. Concerns that the work is lacking in novelty, impact, or interest should not be taken into account. Please visit our website for more information about PLOS ONE.

Reviewer #2: Dear Author,

I’ve reviewed the manuscript and would like to prepare a concise and constructive revision report. Here are the suggested changes for improving the manuscript:

1. Title and Abstract:

The title is appropriate but could be slightly revised to better highlight the significance of care pathways in HFrEF treatment. Consider rephrasing it to something like: "Impact of Clinical Pathways on Enhancing Compliance with Evidence-Based Therapies for HFrEF."

The abstract clearly summarizes the study's purpose, methods, results, and conclusion. However, emphasize the significance of the findings in terms of clinical application and future implications in the conclusion.

2. Introduction:

Add a brief sentence explaining why compliance with GDMT is specifically challenging in clinical practice.

Mention the gap in literature this study addresses more explicitly to enhance its relevance.

3. Methods:

Include more detail on how the care pathway was designed and the rationale for selecting its components.

Provide a brief explanation of the statistical methods used, ensuring clarity for readers without a deep statistical background.

4. Results:

Expand the discussion on the differences observed in early diuretic use and referrals to CHFM in terms of clinical outcomes and patient benefits.

While the use of GDMT was mentioned, elaborate on why it did not reach statistical significance and its implications for the study findings.

5. Discussion:

Include a brief acknowledgment of potential biases introduced by the voluntary use of pathways.

Highlight the broader implications of implementing similar care pathways in other healthcare settings.

6. Conclusion:

Emphasize the study's contribution to the field and outline specific next steps for research or implementation.

General Improvements:

Ensure all abbreviations (e.g., CHFM, GDMT) are clearly defined upon first use.

Check for consistency in formatting tables and figures for better readability.

Reviewer #3: Tables are not seen in the manuscript. Kindly include all the three tables for review.

The article is well written and stresses the importance of diuretics in early stages of heart failure.

The number of people on extra medications like digoxin can be mentioned in the manuscript.

Reviewer #4: Re Manuscript Number PONE-D-24-34453: The authors have presented the results of care pathways for heart failure with reduced ejection fraction within the context of healthcare systems in the United States. The call to action for adhering to guidelines in the hospital setting is admirable. However, the authors should provide more context regarding the low uptake of their intervention in light of more recent evidence (10.1016/j.jacc.2023.12.024). Moreover, elaborating on what distinguishes this study from others in literature (e.g., 10.5837/bjc.2022.005; 10.1002/ehf2.14911) would be beneficial. Further comments are provided below.

Major comments:

1) As suggested by PLOS One journal, I would encourage the authors to follow the STROBE reporting guideline in their manuscript (https://www.strobe-statement.org/).

2) Although not explicitly stated, the theme underlying the arguments presented by the authors in both abstract and introduction sections tends to switch between inpatient and outpatient settings. It is important that authors transparently present arguments relevant only to hospital settings; otherwise, they may risk confusing readers.

3) Unfortunately, tables were not available for review; however, based on what has been stated in text, I would encourage authors not to compare baseline characteristics of participants via statistical measures as this would not provide clinically meaningful insights. Furthermore, considering a large sample size and assuming a high event rate, I argue that using odds ratios could yield exaggerated results—this is also evident from wide precision margins accompanying effect sizes—a risk difference might provide more meaningful interpretation of findings.

4) Authors tend to dichotomize results based on statistical significance; I discourage this approach and invite interpreting results based on clinical significance instead.

Minor comment:

1) Page 6, Line 123: I would suggest that authors specify which guidelines they refer to at this point.

Reviewer #5: I would like to thank the authors for their contribution and the manuscript. The study indeed is significant and it’s encountered commonly in clinical practice due to failure to achieve GDMT in heart failure.

Reviewer #6: References need to be rewritten according to the journal's format.

Some abbreviations need clarification.

Images and tables are missing.

Please rewrite them with images and tables to make the idea clear to the reader.

**Do you want your identity to be public for this peer review?** For information about this choice, including consent withdrawal, please see our Privacy Policy

Reviewer #1: No

Reviewer #2: **Yes: ** Umit Yasar Sinan

Reviewer #3: **Yes: ** Mark Christopher Arokiaraj

Reviewer #4: No

Reviewer #5: No

Reviewer #6: **Yes: ** YAHYA A. A. ABODEA

---

## [Author Response · Author response to Decision Letter 1]

3 Jun 2025

Dear Editor

Dyrad, where in my data hosted provided this link that I can provide to reviwers :

http://datadryad.org/share/oe_qWDYBI6k-3fjv2180cUhbRVkh366O9nd2SKHri5E

but they have instructions : "Link to an anonymous download of your unpublished files. This is a temporary URL and should not be published or made publicly available."

---

## [Decision Letter · Decision Letter 1]

29 Jul 2025

Impact of clinical pathways on enhancing compliance with evidence-based therapies for Heart failure with reduced ejection fraction – A retrospective cohort study

PONE-D-24-34453R1

Dear Dr. Kanaparthy,

We’re pleased to inform you that your manuscript has been judged scientifically suitable for publication and will be formally accepted for publication once it meets all outstanding technical requirements.

Kind regards,

Gbolahan Olatunji, M.D, MPH

Academic Editor

PLOS ONE

Additional Editor Comments (optional):

Reviewers' comments:

Reviewer's Responses to Questions

**Comments to the Author**

Reviewer #2: All comments have been addressed

Reviewer #3: All comments have been addressed

2. Is the manuscript technically sound, and do the data support the conclusions?

Reviewer #2: Yes

Reviewer #3: Yes

3. Has the statistical analysis been performed appropriately and rigorously?

Reviewer #2: Yes

Reviewer #3: Yes

4. Have the authors made all data underlying the findings in their manuscript fully available?

Reviewer #2: Yes

Reviewer #3: Yes

5. Is the manuscript presented in an intelligible fashion and written in standard English?

Reviewer #2: Yes

Reviewer #3: Yes

Reviewer #2: Dear Authors,

Thank you for submitting the revised version of your manuscript entitled:

“Impact of clinical pathways on enhancing compliance with evidence-based therapies for heart failure with reduced ejection fraction – A retrospective cohort study.”

We appreciate the thoughtful and comprehensive way you addressed the reviewer comments. In particular, you have strengthened the discussion regarding the low utilization of the pathway, acknowledged potential selection bias, clarified the definition and clinical interpretation of GDMT adherence, and polished the language throughout the manuscript.

The revisions have significantly improved the clarity, rigor, and overall quality of the manuscript. Your study provides important and practical insights into improving adherence to guideline-directed medical therapy in HFrEF through EHR-embedded pathways, which we believe will be of great interest to the readership.

Best regards,

Reviewer #3: The revision is adequate and all comments have been addressed. in the future more number of patients can be recruited for more comprehensive assessment.

**Do you want your identity to be public for this peer review?** For information about this choice, including consent withdrawal, please see our Privacy Policy

Reviewer #2: **Yes: ** Umit Yasar Sinan

Reviewer #3: **Yes: ** Mark Christopher Arokiaraj

---

## [Editor Report · Acceptance letter]

PONE-D-24-34453R1

PLOS ONE

Dear Dr. Kanaparthy,

I'm pleased to inform you that your manuscript has been deemed suitable for publication in PLOS ONE. Congratulations! Your manuscript is now being handed over to our production team.

Kind regards,

on behalf of

Dr. Gbolahan Deji Olatunji

Academic Editor

PLOS ONE